# Cross-Domain Adaptation of a Chest CT Vision-Language Model for Intracranial Hemorrhage Detection

**Hyeon Jun Park**[1]                                              GUSWNS0429@KHU.AC.KR
**Tae-Seong Kim**[2]                                                    TSKIM@KHU.AC.KR
**Won Hee Lee**[*1,3]                                                   WHLEE@KHU.AC.KR

[1] *Department of Artificial Intelligence, Kyung Hee University, Yongin, Korea*

[2] *Department of Biomedical Engineering, Kyung Hee University, Yongin, Korea*

[3] *Department of Software Convergence, Kyung Hee University, Yongin, Korea*

## Abstract

Pre-trained medical vision-language models (VLMs) offer strong representational capacity, yet their transferability across anatomically distinct CT domains remains underexplored. We investigate cross-domain adaptation of CT-CLIP, a VLM pre-trained on chest CT-report pairs, for intracranial hemorrhage detection in brain CT using the CQ500 dataset. We compare three preprocessing window settings (brain, subdural, and bone) and evaluate a spectrum of adaptation strategies: zero-shot inference, vocabulary fine-tuning, linear probing, LoRA, partial fine-tuning, and few-shot learning. Performance is assessed via AUROC and average precision, with F1-score reported using Youden's J-based thresholds. Zero-shot transfer yields near-chance performance (AUROC 0.48–0.55), confirming a substantial domain gap, while supervised adaptation yields consistent gains with partial fine-tuning achieving the highest AUROC (0.736). In the few-shot setting, classifier-based methods with VocabFine outperform retrieval-based approaches, with performance scaling steadily with shot count. Preprocessing window choice critically influences cross-domain performance, suggesting it should be treated as a core adaptation decision rather than a fixed implementation detail. These findings offer practical guidance for deploying pre-trained medical VLMs beyond their original training domain.

**Keywords:** Vision-Language Model, Domain Adaptation, Intracranial Hemorrhage, CT

## 1. Introduction

Intracranial hemorrhage (ICH) is a life-threatening emergency where rapid, accurate diagnosis is critical. Computed tomography (CT) remains the standard imaging modality in emergency settings due to its rapid acquisition and high diagnostic sensitivity (Sporns et al., 2021). While supervised deep learning models have achieved strong ICH detection performance, they depend heavily on large annotated datasets that are costly to obtain.

Vision-language models (VLMs) pre-trained on image-report pairs offer a label-efficient alternative. CT-CLIP is a CT-focused contrastive language-image pretraining framework designed for broad application without task-specific training, outperforming fully supervised models in multi-abnormality detection (Hamamci et al., 2024). However, its training domain is exclusively chest CT, and transferring it to brain CT introduces a substantial anatomical and intensity distribution shift. Models trained on one imaging domain and directly applied to another suffer dramatic performance degradation without explicit adaptation (Wang et al., 2022).

---

[*] Corresponding author

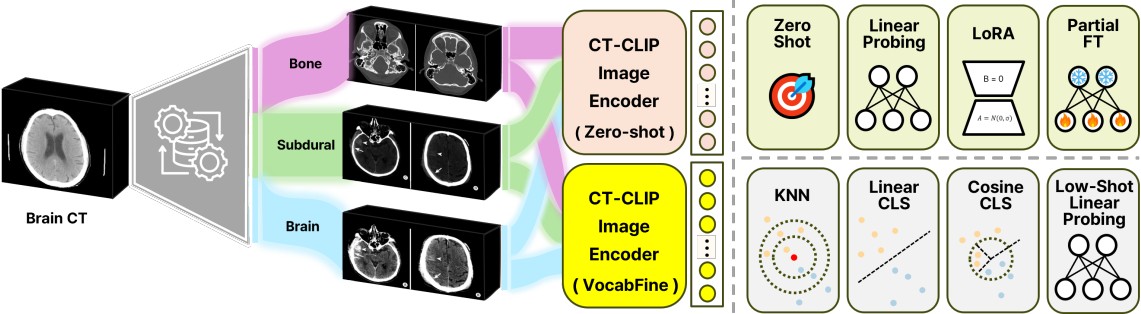

Figure 1: Adaptation strategies evaluated under full-data and few-shot settings.

To address this, we evaluate a spectrum of lightweight adaptation strategies — zero-shot inference, vocabulary fine-tuning, linear probing, partial fine-tuning, LoRA (Hu et al., 2022), and few-shot adaptation — applied to CT-CLIP for ICH detection on the CQ500 dataset. We further quantify how preprocessing window settings (brain, subdural, bone) modulate cross-domain generalization — an understudied but clinically relevant factor. Together, our results offer practical guidance for deploying pre-trained medical VLMs beyond their original training distribution.

## 2. Methods

**Dataset and Image Preprocessing** We used the CQ500 dataset for training and evaluation. DICOM scans were converted to HU-preserving NIfTI volumes and resampled to 1 mm isotropic resolution. Three windowing settings were applied: brain (WL/WW: 40/80), subdural (80/200), and bone (600/2800) (Wang et al., 2021), enabling systematic comparison of preprocessing choices on cross-domain transfer performance.

**Adaptation Strategies** We evaluated two backbone starting points: CT-CLIP and VocabFine, a variant with additional medical terminology alignment (Wortsman et al., 2022). Adaptation strategies were compared under both full-data and few-shot regimes (Figure 1). In the full-data setting, we compared four strategies. *Zero-shot* performs direct inference without any adaptation. *Linear Probing (LiPro)* trains a supervised linear classifier on frozen backbone features. *LoRA* applies parameter-efficient low-rank adaptation to the pre-trained backbone (Hu et al., 2022). *Partial Fine-Tuning (FT)* updates the lower 50% of layers in both the spatial and temporal transformer blocks of the image encoder, while keeping the remaining layers frozen.

In the few-shot setting (brain window only), we evaluated four strategies using 1, 5, 10, and 20 shots per class, averaged over 10 repeated samplings. *KNN* predicts labels via nearest-neighbor retrieval in frozen feature space. *LinearCLS* and *CosineCLS* (Chen et al., 2019) train linear and cosine-similarity classifiers on fixed features, respectively. *Low-shot LiPro* applies the standard linear probing pipeline to the few-shot support set.

**Evaluation** Performance was assessed using the area under the receiver operating characteristic curve (AUROC), average precision (AP), and F1-score. AP was additionally emphasized in the few-shot setting, where F1-score can be sensitive to threshold selection under limited validation data. Decision thresholds for F1-score were determined on the validation set via Youden's J statistic and applied to the test set (Youden, 1950).

Table 1: Full-data ICH detection results on CQ500. **Bold** indicates the best value per column.

| Backbone | Window | Zero-shot AUROC | Zero-shot F1 | LiPro AUROC | LiPro F1 | LoRA AUROC | LoRA F1 | Partial FT AUROC | Partial FT F1 |
|---|---|---|---|---|---|---|---|---|---|
| CT-CLIP | Brain | 0.529 | 0.515 | 0.569 | 0.525 | 0.574 | **0.552** | 0.610 | 0.519 |
| CT-CLIP | Subdural | 0.528 | 0.500 | 0.571 | **0.579** | 0.576 | 0.543 | 0.606 | 0.485 |
| CT-CLIP | Bone | **0.550** | **0.566** | 0.573 | 0.543 | 0.574 | 0.549 | 0.618 | 0.282 |
| VocabFine | Brain | 0.489 | 0.128 | **0.628** | 0.444 | **0.625** | 0.455 | **0.736** | 0.595 |
| VocabFine | Subdural | 0.477 | 0.163 | 0.613 | 0.367 | 0.610 | 0.506 | 0.727 | **0.597** |
| VocabFine | Bone | 0.485 | 0.196 | 0.579 | 0.418 | 0.576 | 0.423 | 0.713 | 0.523 |

Table 2: Few-shot ICH detection results on CQ500 (brain window). **Bold** indicates the highest mean value per column. Values are reported as mean ± standard deviation.

| Method | Backbone | 1-shot AUROC | 1-shot AP | 5-shot AUROC | 5-shot AP | 10-shot AUROC | 10-shot AP | 20-shot AUROC | 20-shot AP |
|---|---|---|---|---|---|---|---|---|---|
| KNN | CT-CLIP | 0.519±0.037 | 0.425±0.021 | 0.546±0.067 | 0.450±0.038 | 0.568±0.062 | 0.471±0.047 | 0.563±0.066 | 0.457±0.049 |
| KNN | VocabFine | 0.540±0.046 | 0.439±0.026 | 0.560±0.035 | 0.455±0.025 | 0.569±0.054 | 0.468±0.045 | 0.578±0.042 | 0.480±0.038 |
| LinearCLS | CT-CLIP | 0.533±0.051 | 0.490±0.058 | 0.541±0.070 | 0.489±0.069 | 0.546±0.074 | 0.499±0.075 | 0.587±0.027 | 0.509±0.051 |
| LinearCLS | VocabFine | 0.567±0.078 | 0.514±0.057 | 0.576±0.091 | 0.527±0.068 | **0.577**±0.078 | **0.513**±0.066 | **0.623**±0.018 | 0.553±0.022 |
| CosineCLS | CT-CLIP | 0.523±0.059 | 0.490±0.069 | 0.530±0.074 | 0.480±0.070 | 0.551±0.069 | 0.499±0.070 | 0.593±0.031 | 0.515±0.023 |
| CosineCLS | VocabFine | **0.581**±0.080 | **0.520**±0.057 | **0.581**±0.085 | **0.534**±0.070 | 0.568±0.091 | 0.507±0.075 | 0.619±0.032 | **0.556**±0.037 |
| Low-shot LiPro | CT-CLIP | 0.530±0.004 | 0.450±0.006 | 0.543±0.036 | 0.467±0.052 | 0.549±0.057 | 0.479±0.058 | 0.579±0.030 | 0.484±0.041 |
| Low-shot LiPro | VocabFine | 0.567±0.043 | 0.493±0.035 | 0.558±0.086 | 0.501±0.060 | 0.568±0.075 | 0.498±0.068 | 0.616±0.011 | 0.542±0.016 |

## 3. Results and Discussion

Table 1 shows that zero-shot transfer from chest to brain CT yields near-chance performance (AUROC 0.48–0.55), confirming a substantial domain gap. Supervised adaptation consistently improves performance across all strategies, with Partial Fine-Tuning achieving the highest AUROC (0.736, VocabFine + Brain window). VocabFine generally outperforms CT-CLIP under supervised adaptation, while CT-CLIP proves more robust in the zero-shot setting, suggesting that vocabulary alignment shifts the model toward task-specific representations at the cost of general transferability. Preprocessing window choice has a meaningful impact: the subdural window tends to yield stronger F1 scores, while the bone window produces unstable F1 despite competitive AUROC, indicating poor calibration. These findings suggest that window selection should be treated as a core adaptation decision rather than a fixed preprocessing step.

In the few-shot setting (Table 2), performance increases steadily with shot count, with VocabFine consistently outperforming CT-CLIP. CosineCLS with VocabFine achieves the strongest early performance (AUROC 0.581 at 1- and 5-shot), while LinearCLS with VocabFine leads at 20-shot (AUROC 0.623). KNN and Low-shot LiPro lag behind classifier-based methods, suggesting that even a minimal linear boundary leverages few labeled examples more effectively than retrieval. Overall, our results demonstrate that pre-trained medical VLMs can be meaningfully adapted to new CT domains with limited supervision, and that the preprocessing strategy plays an equally important role in cross-domain generalization.

**Acknowledgments** This work was supported in part by the Institute of Information & Communications Technology Planning & Evaluation (IITP), funded by the Korea government (MSIT), under grants RS-2024-00509257 (Global AI Frontier Lab) and IITP-2026-RS-2024-00438239 (ITRC); in part by the Korea Health Technology R&D Project through the

Korea Health Industry Development Institute (KHIDI), funded by the Ministry of Health and Welfare (RS-2025-02293110); and in part by the National Research Foundation of Korea (NRF), funded by MSIT (RS-2026-25485605).

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
