# OpenReview forum: "Cross-Domain Adaptation of a Chest CT Vision-Language Model for Intracranial Hemorrhage Detection"
_MIDL.io/2026/Short_Papers — MIDL 2026 - Short Papers Poster_

### Official Review · Reviewer_Ah96 · 2026-05-02
**Review for chest CT VLM**

**Rating:** 3
**Confidence:** 4

**Review:**

The paper addresses a highly relevant problem: the "domain shift" that occurs when a medical foundation model is applied to an anatomical region outside its training data. The quality of the evaluation is high, covering both full-data and few-shot regimes using the CQ500 dataset. The clarity of the results regarding the impact of preprocessing windows is a major contribution, as it challenges the standard practice of using fixed windowing across all models.  The comparison between standard CT-CLIP and the "VocabFine" variant provides original insights into how medical terminology alignment affects general transferability. However, the absolute performance levels suggest that even the best-adapted models struggle to match the accuracy of specialized, fully-supervised brain CT models. The paper is significant for its practical guidance on deploying VLMs in data-constrained clinical environments.

**Summary:**

This research investigates the transferability of CT-CLIP, a vision-language model (VLM) trained on chest CT scans, to the anatomically distinct domain of brain CT for intracranial hemorrhage (ICH) detection. Utilizing the CQ500 dataset, the authors compare various adaptation techniques, such as partial fine-tuning, LoRA, and few-shot learning, across different Hounsfield Unit windowing settings. The results highlight a significant domain gap where zero-shot performance is near-chance, but supervised adaptation, particularly partial fine-tuning, yields substantial improvements. Ultimately, the study underscores that preprocessing window selection is a critical factor in the cross-domain generalization of medical VLM

**Strengths:**

The paper's strength lies in its systematic comparison of parameter-efficient adaptation methods and its focus on clinically relevant preprocessing factors. The discovery that partial fine-tuning of 50% of the encoder layers consistently outperforms LoRA provides actionable advice for researchers. The few-shot analysis also demonstrates how performance scales from 1 to 20 shots, which is crucial for rare abnormality detection.

**Weaknesses:**

A notable weakness is the low baseline zero-shot performance, which indicates a massive anatomical gap that the model cannot overcome without significant labeled data. Furthermore, the study is limited to a single external dataset, and the maximum AUROC achieved (0.736) remains relatively modest compared to SOTA supervised models. There is also no detailed analysis of specific hemorrhage types, such as distinguishing between epidural and intraparenchymal bleeds

**Justification Of Rating:**

This is a solid benchmarking study that provides essential insights into the limitations of current CT foundation models for cross-domain tasks. While the methodological novelty is limited to the systematic evaluation of existing techniques, the focus on windowing as an adaptation hyperparameter is valuable enough for a borderline accept in the short paper track.

---

### Decision · Program_Chairs · 2026-05-08

Accept (Poster)